# Thermal Stability and Antioxidant Activity of Bioactive Compounds in Bread Enriched with Bee Pollen and Bee Bread

**DOI:** 10.3390/antiox12091691

**Published:** 2023-08-30

**Authors:** Seymanur Ertosun, Volkan Aylanc, Soraia I. Falcão, Miguel Vilas-Boas

**Affiliations:** 1Centro de Investigação de Montanha (CIMO), Instituto Politécnico de Bragança, Campus de Santa Apolónia, 5300-253 Bragança, Portugal; seymanur@ipb.pt (S.E.); volkan@ipb.pt (V.A.); 2Laboratório Associado para a Sustentabilidade e Tecnologia em Regiões de Montanha (SusTEC), Instituto Politécnico de Bragança, Campus de Santa Apolónia, 5300-253 Bragança, Portugal; 3Departamento de Química e Bioquímica, Faculdade de Ciências, LAQV-REQUIMTE, Universidade do Porto, 4169-007 Porto, Portugal

**Keywords:** bee products, functional food, phenolic compounds, phenolamides, natural antioxidants

## Abstract

Bee pollen (BP) and bee bread (BB) are natural food sources containing a wide variety of bioactive compounds, complementing their rich nutritional composition. These bee products are being explored to empower functional foods, with the term functionality being dependent on the bioactive compounds added to the food matrix. However, there is not enough evidence of the effect of heat on these compounds during food processing and production and how it impacts their biological activity. Here, we enriched traditional bread by adding BP and BB at different proportions of 1 to 5% and tested the thermal stability of their bioactive compounds through several spectroscopic and chromatographic analyses. Adding bee pollen and bee bread to bread resulted in a 4 and 5-fold increase in total phenolic content, respectively. While not all the 38 phenolic and phenolamide compounds identified in the raw BP and BB were detected in the processed bread, phenolamides were found to be more resilient to baking and heat treatment than flavonoids. Still, the enriched bread’s antioxidant activity improved with the addition of BP and BB. Therefore, incorporating bee products into heat-treated products could enhance the functionality of staple foods and increase the accessibility to these natural products.

## 1. Introduction

Currently, one of the most important issues in preventing diseases and living a healthier life is healthy consumption with the idea of ‘‘You are what you eat’’ [1]. Although the requirements and perception of food nutrition from 2500 years ago are expected to have evolved significantly, this principle resembles an old statement attributed to Hippocrates “Let food be thy medicine and medicine be thy food” [2]. Food has become a marketable product that promises more than just survival, and people expect more efficiency from the products they consume with this belief [3]. In this context, ordinary foods have begun to be enriched with different products in many ways and made more functional with the demand for a better quality of life.

The idea of adding phytochemicals to foods is quite logical as it aims to increase the benefits. Phytochemicals, abundantly present in plants and plant products, are reported to have the effect of protecting cells from oxidative stress [4], anti-tumour activity [5], strong inhibition activity on cancer derivatives [6], anti-inflammatory activity [7] and protection against microorganisms [8], among many other functionalities. For instance, lycopene intake in rats was found to decrease prostate cancer [9], naringenin was used for ulcerative colitis and decreased inflammation [10], and anthocyanins were tested and found to reduce alcohol-induced liver damage [11]. 

This drives consumers to search for natural products that are both health-promoters and high in nutritional value, and bee pollen and bee bread are potential natural sources that meet those needs. Bees collect pollen from different kinds of flowers and aggregate them with their own secretions into pellets, which are transported in the rear legs to the hive. This product is called BP [12]. When BP is stored in honeycomb cells, some honey and honeybee secretions are mixed, transforming it into BB, which is a naturally fermented product [13], as shown in Figure 1. The bioactive compound composition of BP and BB is mainly flavonoids (naringenin, isorhamnetin, quercetin, kaempferol and rutin), phenolic acids (caffeic acid, cinnamic acid and ferulic acid) and phenolamides, depending on their plant origin [13].

Functionalization of traditional food products with the addition of bee products such as honey, propolis, BP, and BB has been increasing in recent years [2]. In a study conducted by Conte et al. [14], gluten-free bread fortified with BP showed an increase in the overall phenolic content, which implies great antioxidant activity of enriched bread. Another study on BP-enriched biscuits found a similar increase in the bioactive compounds, besides sugar, protein, ash and fiber, leading the authors to propose biscuits as a good vehicle for the uptake of pollen in the human diet [15]. Moreover, it has been reported that bee products both increase the nutritional value of products such as yoghurt [16], cookies [17] and sausage [18] and provide protection against food contaminants [8]. Not only BP and BB but also other bee products like honey, propolis or royal jelly have been used together with various food products such as meat, beverages, bakery and dairy products to increase their nutritional values and biological activity, as well as to extend their shelf life [19,20,21,22]. For example, adding honey to yoghurt increased the nutritional value of yoghurt, and *Bifidobacteria* were activated by mixing honey with yoghurt milk, meaning that honey could be used as both a sweetener and a prebiotic enhancer in yoghurt production [23]. In another study, mixing propolis with fish burgers both increased antioxidant properties and improved sensory properties [24].

In our previous work [25], the addition of BP and BB was evaluated for their impact on the design of functionalized bread. The characteristics of the enriched breads were tested in different parameters such as texture, color, sensorial perception or microbial stability, along with their nutritional value. However, it remains unknown if the addition of those bee products in bread may have an effective impact on bioactivity, particularly if bioactive compounds can resist the baking processes and the required thermal conditions.

Therefore, this study aims to clarify the impact of the baking processes on the bioactivity of bread enriched with bee pollen and bee bread and contribute to the development of an improved daily product for better efficacy in continuous consumption of bioactive compounds.

## 2. Materials and Methods

### 2.1. Chemicals and Reagents

Ethanol, methanol, sodium phosphate, potassium phosphate, potassium ferrocyanide, trichloroacetic acid, acetonitrile, formic acid, sodium hydroxide and gallic acid were purchased from Fisher Scientific (Pittsburgh, PA, USA). Folin–Ciocâlteu’s reagent was purchased from Panreac Applichem (Barcelona, Spain). Iron(III) chloride and aluminium chloride were purchased from Acros Organics (Pittsburgh, PA, USA). 2,2-diphenyl-1-picrylhydrazyl (DPPH), quercetin, *p*-coumaric acid and chrysin were purchased from Sigma-Aldrich (St. Louis, MO, USA). Kaempferol was purchased from Extrasynthese (Genay, France). Water was treated in a Milli-Q water-purification system (TGI pure system, Houston, TX, USA).

### 2.2. Bee Pollen and Bee Bread Samples

BP and BB samples were collected in Bragança, northeast region of Portugal, during the spring of 2019, from *Apis mellifera iberiensis* hives. BP was collected through BP traps placed in front of the hive, Figure 1, while BB was removed from the honeycombs manually. Both BP and BB, once collected, were crushed, homogenized, lyophilized and maintained in a desiccator until further analyses. The botanical origin of each sample was confirmed in a previous study [25], in which both samples were classified as multifloral, with *Cytisus striatus* and *Crepis capillaris* being revealed as the most representative plants in the BP sample (21% and 17%, respectively), while *Castanea sativa* and *Rubus* sp. were found in the BB sample, at 39% and 18%, respectively.

### 2.3. Bread Preparation

The bread was prepared using a home-making bread machine (Tefal bread maker XXL, Windsor, Berkshire, UK). The selected bread-making program included dough preparation (140 min) and baking (20 min). After baking, the bread was cooled at room temperature for 2 h before further analyses. All the ingredients were provided by a local bakery Nopabril LDA, from Bragança, Portugal (wheat flour and yeast were from Lallemand, Setúbal, Portugal). Breads were prepared following the bakery’s recipe; the formulations used to prepare the samples are shown in Table 1. Three different bread formulations were produced for each type of bee product with the addition of different proportions (1%, 3% and 5%) of supplementation flour basis. A bread without the fortification of BP or BB was used as control.

### 2.4. Phenolic Compound Extraction

The extraction was performed according to a previously described method [13]. Briefly, 2 g of the sample (raw bee products or bread) were mixed with 40 mL of 80% ethanol/water and kept under magnetic stirring at room temperature for 6 h. The resulting mixture was filtered and the residue was re-extracted in the same conditions. After the second extraction, the solutions were combined, first evaporated at 40 °C in a rotavapor (Rotary Evaporator model Hei-VAP from Heidolph, Schwabach, Germany) then lyophilized (FreeZone 4.5, Labconco, Kansas City, MO, USA), and finally stored at −20 °C until further analysis.

### 2.5. Spectrophotometric Determination of Phenolic Compounds

#### 2.5.1. Total Phenolic Content

The total phenolic content was determined using the Folin–Ciocâteau method [13]. An aliquot of 0.5 mL of ethanolic extract (1 mg/mL) of BP, BB or bread was mixed with 0.25 mL of Folin–Ciocâlteu reagent. After 3 min, 1 mL of 20% sodium carbonate was added and the final volume was set to 5 mL with deionized water. The mixture was left in a water bath at 70 °C for 10 min and then cooled for 30 min. The absorbance was read at 760 nm using a spectrophotometer (Analytikijena 200–2004 spectrophotometer from Analytik Jena, Jena, Germany). The phenolic content of samples was expressed as milligrams of gallic acid equivalent per gram of dry weight sample (mg GAE/g).

#### 2.5.2. Total Flavonoid Content 

The flavonoid content was recorded spectrophotometrically according to a previously reported method [13]. For this, 0.2 mL of ethanolic extract (5 mg/mL) of BP, BB or bread was blended with 0.2 mL of AlCl_3_ (2% AlCl_3_, in 5% glacial acetic acid). Then, 2.8 mL of 5% acetic acid/methanol was added to the mixture. The absorbance was read at 415 nm using a spectrophotometer after 30 min. The flavonoid content of samples was expressed as milligrams of quercetin equivalent per gram of dry weight sample (mg QE/g).

### 2.6. LS-DAD-ESI-MS^n^ Analysis

For profile analysis of the bioactive compounds, 20 mg of the previous extracted sample (see Section 2.4, Phenolic Compound Extraction) was taken and dissolved in ethanol/water (80:20) to reach a concentration of 10 mg/mL. The same procedure was applied for control bread and bread incorporating BP or BB. The samples were filtered through a 0.22 μm membrane filter and kept at −20 °C until analysis. 

A Dionex UltiMate 3000 ultra-pressure liquid chromatography instrument connected to a diode array and attached to a mass detector was used for LC/DAD/ESI-MS^n^ analyses (Thermo Fisher Scientific, San Jose, CA, USA). The analysis was conducted on a Macherey-Nagel Nucleosil C18 column (250 mm × 4 mm id; particle diameter of 5 mm, end-capped), with temperature kept constant at 30 °C. The liquid chromatography conditions applied were based on previous work [12], with a flow rate set to 1 mL/min and 10 μL as the injection volume. The final spectra data were accumulated in the wavelength interval of 190–600 nm. 

The LTQ XL linear ion trap mass spectrometer (Thermo Fisher Scientific, San Jose, CA, USA) equipped with an ESI source was set in the negative ion mode with the following parameters: source voltage 5 kV; tube lens voltage, −20 V; capillary voltage, −65 V; capillary temperature, 325 °C; and sheath and auxiliary gas flow (N2) 50 and 10 (arbitrary units), respectively.

Mass spectra were acquired in a full range of 100–1000 *m*/*z*. For the fragmentation study, a data-dependent scan was performed by deploying collision-induced dissociation (CID). The normalized collision energy of the CID cell was set at 35 (arbitrary units). All data acquisition was performed using the Xcalibur^®^ software, version 4.2 (Thermo Fisher Scientific, San Jose, CA, USA). The elucidation of the phenolic compounds was achieved by comparison of their chromatographic behavior, UV spectra and MS information to those of reference compounds. When standards were not available, the structural information was confirmed with UV data combined with MS fragmentation patterns previously reported in the literature. 

Quantification was achieved using calibration curves for *p*-coumaric acid (0.00925–0.4 mg/mL; y = 1.9 × 10^7^x − 12,927; R^2^ = 9.957), quercetin (0.037–1.6 mg/mL; y = 4 × 10^6^x − 10,216; R^2^ = 9.970), kaempferol (0.037–1.6 mg/mL; y = 4.3 × 10^6^x − 13,567; R^2^ = 9.981), chrysin (0.0185–0.8 mg/mL; y = 1.2 × 10^7^x − 51,265; R^2^ = 9.999) and naringenin (0.0185–0.8 mg/mL; y = 8 × 10^6^x − 10,998; R^2^ = 9.976). All compounds were quantified using the calibration curve of the structurally closest standard, and the final result was given in equivalent terms. Each value resulted from three different assays and is expressed as mg/g of sample.

### 2.7. Antioxidant Activity

#### 2.7.1. DPPH Free-Radical Scavenging Activity

DPPH free-radical scavenging activities of samples (BP, BB and bread) were performed according to Brand-Williams et al. [26] with slight modification. An aliquot of 0.15 mL of sample extract (0.03–0.43 mg/mL) was mixed with 0.15 mL of DPPH solution (50 mg/L) and the absorbance was read at 515 nm using an ELX800 Microplate Reader (Bio-Tek Instruments, Inc., Winooski, VT, USA). The percentage of radical inhibition was calculated using the following equation:$$\%Inhibition=[(A_{DPPH}-A_{Sample})/A_{DPPH}]\times 100$$

The amount of antioxidants necessary to decrease the initial DPPH concentration by 50% (EC_50_) was achieved by plotting the inhibition percentage against the extract concentration (dry weight).

#### 2.7.2. Reducing Power

The reducing power assay was performed according to the previously described method [27]. A volume of 0.25 mL (1 mg/mL) of the sample (BP, BB and bread) extract was mixed with 1.25 mL of phosphate buffer (0.2 M, pH 6.6) and 1.25 mL of 1% potassium ferricyanide. The solution was left in a water bath at 50 °C for 20 min. Then, 1.25 mL of 10% trichloroacetic acid was added to the mixture and centrifuged at 3000 rpm for 10 min. From the upper layer, 1.25 mL was mixed with 1.25 mL of deionized water and 0.25 mL of 0.1% FeCl_3_. The absorbance was read at 700 nm and the results were expressed as milligrams of gallic acid equivalent per gram of dry weight sample (mg GAE/g).

### 2.8. Statistical Analysis

All analyses were performed in triplicate and the results were denoted as mean ± standard deviation (SD). The obtained results from calculations were analyzed using GraphPad Prism version 8 (San Diego, CA, USA). *p* < 0.05 was considered significant.

## 3. Results and Discussion

### 3.1. Total Phenolic and Flavonoid Content 

To assess the temperature impact on the phenolic and flavonoid content of fortified bread, we first checked the total amount of these compounds in the raw BP and BB. This allowed us to know their phenolic and flavonoid composition, and the expected incorporation within the bread, as seen in Figure 2.

The results showed that the phenolic content in raw BB was approximately two times higher than that in BP, as shown in Figure 3a. On the other hand, BP was significantly richer in flavonoids (*p* < 0.05) than BB, as shown in Figure 3b. The differences observed for those two classes of compounds are most probably related to the environmental conditions [28] and particularly the differences between the plant origin of the samples, as mentioned in Section 2.2. Nevertheless, other factors such as the lactic fermentation that BB undergoes during maturation, the harvesting season, and processing and storage conditions of the bee products, may be additional factors that could impact the phenolic composition [29]. Nonetheless, the observed levels are consistent with the values reported in the literature [30,31].

Regarding the enriched bread, the phenolic content of the bread increased with the amount of bee product added, as shown in Figure 3c. These changes resemble the phenolic differences observed for the raw material, which explains why the BB addition results in higher values. A similar trend was reported for other foods fortified with BP, i.e., higher phenolic content relative to the amount of BP added [14,15]. In terms of flavonoids, there is also an increment in the content with the amount of BP and BB added in the baking procedure. However, although raw BP showed higher flavonoid content compared to BB, it was significantly lower when mixed with bread, especially in samples with 1% and 3%, as shown in Figure 3d. This may have occurred due to the inability to provide a sufficiently homogeneous distribution of BB in the bread loaf or to a non-homogeneous incorporation with flour.

These patterns provide evidence for the stability of compounds during the bread-making process, because, if the temperature had a significant effect on the bioactive compounds of BP and BB, it would be most likely to cause the degradation [32] of the compounds and it would be more difficult to quantify them and see a pattern with an increasing trend. The second piece of evidence is that the phenolic compound content of the samples was higher in all cases compared to the control group. Furthermore, both BP and BB consist of pollen grains, and these pollen grains have a double-layered pollen wall, called exine and intine [33]. Exine can exhibit thermal stability above 400 °C [34], which provides considerable protection for the bioactive compounds of both bee products.

### 3.2. LC/DAD/ESI-MS^n^ Bioactive Compound Analysis

The bioactive compound composition of raw BP, BB and enriched bread samples was characterized by LC/DAD/ESI-MS^n^, in the negative ion mode. The chromatographic profile, recorded at 280 nm, allowed the identification of 38 bioactive compounds in raw BP and BB, comprising 18 phenolic and 20 phenolamide compounds, Table 2. Previous studies described both phenolic compounds and phenolamides within the bioactive composition of these bee products [12,13]. Within the phenolics, flavonol derivatives such as quercetin, kaempferol, isorhamnetin and herbacetin glycosides were the main compounds found for both bee products. On the other hand, phenolamides were detected in higher concentrations, particularly spermidine and spermine derivatives.

Phenolamides are plant secondary metabolites that can be found in different parts of a plant, such as pollen and seeds, and are associated with diverse functions such as protection against environmental stresses, sporopollenin formation, pollen protection against UV or pollination [35]. Hydroxycinnamic acids like *p*-coumaroyl, caffeoyl and feruloyl can conjugate with phenolamine compounds [36]. The formation of the amide linkage between a phenolamide and phenolic acid can occur at different positions: *N^1^*, *N^5^* and *N^10^* positions [37]. Previously, it was found that phenolamides are abundant compounds within Asteraceae, Fabaceae and Rosaceae plant families, the main pollen types found in the samples under study [38,39].

**Table 2 antioxidants-12-01691-t002:** Phenolic and phenolamide compounds of the raw BP and BB (mg/g).

Peak	tR _(min)_	λ_max_ (nm)	[M-H]^−^ *m*/*z*	MS^n^ (% Base Peak)	Proposed Compound	BP	BB
1	7.54	257, 353	625	MS^2^: 301 (100), 300 (99), 445 (85), 271 (18)	Quercetin-*O*-diglucoside ^a,e,g^	0.07 ± 0.00	ND
2	8.51	272, 326sh, 353sh	639	MS^2^:271 (10), 300 (34), 315 (91), 459 (100), 477 (11), 624 (20)	Methyl herbacetin-*O*-dihexoside ^a,c,d^	0.16 ± 0.00	0.26 ± 0.00
3	9.68	265, 348	609	MS^2^: 285 (100), 429 (49)	Kaempferol-*O*-dihexoside ^a,d^	0.02 ± 0.00	ND
4	10.78	272, 326sh, 353sh	623	MS^2^: 299 (61), 300 (38), 314 (100), 315 (69), 459 (86), 477 (19)	Methyl herbacetin-*O*-rutinoside ^a,c^	ND	0.04 ± 0.00
5	11.62	255, 353	609	MS^2^: 315 (100), 314 (47), 459 (51), 300 (20)	Isorhamnetin-*O*-pentosyl-hexoside ^a,e^	0.13 ± 0.00	0.07 ± 0.00
6	12.15	266, 349	593	MS^2^: 284 (94), 285 (57), 431 (100), 447 (20)	Kaempferol-3-*O*-rutinoside ^a,b^	0.02 ± 0.00	0.01 ± 0.00
7	12.33	-	623	–	Quercetin derivative ^a^	ND	0.07 ± 0.00
8	12.49	255, 354	563	MS^2^: 519 (100); MS^3^: 315 (100)	Isorhamnetin-3-*O*-malonyl glucoside ^a,g^	0.06 ± 0.00	ND
9	12.59	256, 354	463	MS^2^: 301 (100)	Quercetin-3-*O*-glucoside ^a,b^	0.02 ± 0.00	0.05 ± 0.00
10	13.38	267, 347	593	MS^2^: 285 (100)	Kaempferol-*O*-hexosyl-deoxyhesoside ^a,i^	0.02 ± 0.00	ND
11	13.64	270	477	MS^2^: 315 (100), 462 (42), 300 (14); MS^3^: 300 (100)	Methyl herbacetin-3-*O*-hexoside ^a,c^	ND	0.03 ± 0.00
12	14.1	265, 347	447	MS^2^: 285 (100), 284 (80)	Kaempferol-*O*-hexoside ^a,h^	ND	0.07 ± 0.00
13	14.22	254, 347	447	MS^2^: 301 (100)	Quercetin-3-*O*-rhamnoside ^a,b,e^	0.49 ± 0.00	ND
14	14.37	254, 355	477	MS^2^: 314 (100), 315 (45)	Isorhamnetin-*O*-hexoside ^a,h^	ND	0.04 ± 0.00
15	14.83	277, 311	301	MS^2^: 283 (100), 286 (40)	Hesperetin ^a,b^	0.03 ± 0.00	ND
16	15.04	265, 345	533	MS^2^: 489 (100); MS^3^: 285 (100)	Kaempferol-*O*-malonyl-hexoside ^a,g^	ND	0.04 ± 0.00
17	15.46	255, 355	563	MS^2^: 283 (100), 286 (40); MS^3^: 315 (100)	Isorhamnetin-*O*-malonyl-hexoside ^a,h^	0.01 ± 0.00	ND
18	15.83	295, 315	630	MS^2^: 468 (100), 494 (84), 358 (7); MS^3^: 332 (100)	*N^1^, N^5^, N^10^*-tricaffeoylspermidine ^a,d,f^	ND	0.18 ± 0.00
19	16.22	264, 341	431	MS^2^: 285 (100)	Kaempferol-3-*O*-rhamnoside ^a,c^	0.04 ± 0.00	ND
20	16.73	295, 230	630	MS^2^: 468 (100), 494 (100), 358 (7); MS^3^: 315 (100)	*N^1^*, *N^5^*, *N^10^*-tricaffeoylspermidine (isomer) ^a,d,f^	0.81 ± 0.10	1.24 ± 0.20
21	18.18	295, 311	614	MS^2^: 494 (25), 478 (100), 452 (69), 358 (20)	*N^1^*-*p*-coumaroyl-*N^5^*, *N^10^*-dicaffeoylspermidine ^a,e^	0.07 ± 0.00	0.05 ± 0.00
22	18.27	299, 308	478	MS^2^: 358 (100), 332 (12), 145 (5)	*N^1^*-acetyl-*N^5^*, *N^10^*-di-*p*-coumaroylspermidine ^a^	0.37 ± 0.00	ND
23	18.8	295, 311	614	MS^2^: 478 (100), 468 (20), 452 (68), 342(5)	*N^1^*-*p*-coumaroyl-*N^5^*, *N^10^*-dicaffeoylspermidine (isomer) ^a,e^	ND	0.15 ± 0.00
24	19.46	295, 311	614	MS2: 494 (25), 478 (100), 452 (71), 358 (22)	*N^1^*-*p*-coumaroyl-*N^5^*, *N^10^*-dicaffeoylspermidine (isomer) ^a,e^	1.06 ± 0.00	0.18 ± 0.00
25	20.14	295, 318	644	MS^2^: 358 (8), 482 (75), 508 (100); MS^3^: 332 (27), 358 (100), 372 (49)	*N^1^*-feruloyl-*N^5^*, *N^10^*-dicaffeoylspermidine (isomer) ^a,e^	ND	0.06 ± 0.00
26	20.46	295, 310	598	MS^2^: 478 (46), 462 (100), 452 (46), 342 (14)	*N^1^*, *N^5^*-di-*p*-coumaroyl-*N^10^*-caffeoylspermidine ^a,e^	ND	0.04 ± 0.00
27	21.38	295, 310	582	MS^2^: 462 (100), 436 (9), 342 (7)	*N^1^*, *N^5^*, *N^10^*-tri-*p*-coumaroylspermidine ^a,e^	0.34 ± 0.00	0.42 ± 0.00
28	22.37	294, 309	598	MS^2^: 462 (100), 478 (39), 452 (34), 342 (14)	*N^1^*, *N^5^*-di-*p*-coumaroyl-*N^10^*-caffeoylspermidine (isomer) ^a,e^	0.84 ± 0.00	0.45 ± 0.00
29	22.95	295, 310	582	MS^2^: 462 (100), 436 (9), 342 (7)	*N^1^*, *N^5^*, *N^10^*-tri-*p*-coumaroylspermidine (isomer) ^a,e^	1.96 ± 0.10	0.83 ± 0.00
30	24.34	292, 305	582	MS^2^: 342 (100), 436 (9), 462 (100)	*N^1^*, *N^5^*, *N^10^*-tri-*p*-coumaroylspermidine (isomer) ^a,e^	0.39 ± 0.00	0.76 ± 0.00
31	25.3	295, 310	582	MS^2^: 462 (100), 436 (9), 342 (7)	*N^1^*, *N^5^*, *N^10^*-tri-*p*-coumaroylspermidine (isomer) ^a,e^	ND	0.25 ± 0.00
32	26.70	295, 305	582	MS^2^: 342 (100), 436 (9), 462 (100)	*N^1^*, *N^5^*, *N^10^*-tri-*p*-coumaroylspermidine (isomer) ^a,e^	3.44 ± 0.20	1.90 ± 0.20
33	27.19	270	785	MS^2^: 665 (100), 545 (14), 639 (13); MS^3^: 545 (100)	Tetracoumaroyl spermine ^a,j^	0.29 ± 0.00	ND
34	27.75	295, 308	612	MS^2^: 492 (100); MS^3^: 372 (100), 449 (24)	Feruloyl dicoumaroyl spermidine ^a,j^	0.37 ± 0.00	0.02 ± 0.00
35	28.81	280, 307sh	785	MS^2^: 545 (14), 639 (13), 665 (100); MS^3^: 545 (100), 546 (33)	Tetracoumaroyl spermine (isomer) ^a,j^	2.01 ± 0.10	0.17 ± 0.10
36	30.53	289, 306sh	785	MS^2^: 665 (100), 545 (13), 639 (13); MS^3^: 545 (100)	Tetracoumaroyl spermine (isomer) ^a,j^	0.56 ± 0.10	ND
37	32.19	293, 310	785	MS^2^: 665 (100), 545 (13), 639 (13); MS^3^: 545 (100)	Tetracoumaroyl spermine (isomer) ^a,j^	1.05 ± 0.10	ND
38	34.25	299, 310	785	MS^2^: 665 (100), 545 (13), 639 (13); MS^3^: 545 (100)	Tetracoumaroyl spermine (isomer) ^a,j^	1.26 ± 0.00	0.11 ± 0.00
Total phenolic compounds (mg/g)						1.07	0.61
Total phenolamides (mg/g)						14.82	6.81

Confirmed with ^a^ MS^n^ fragmentation; ^b^ Standard; References: ^c^ [40]; ^d^ [41]; ^e^ [42]; ^f^ [43]; ^g^ [44]; ^h^ [45]; ^i^ [46]; ^j^ [47]. BP: bee pollen and BB: bee bread. Values are expressed as mg of each compound/g sample. ND = not detected. sh: shoulder.

Within the flavonols, quercetin-3-*O*-rhamnoside (*m*/*z* 463), isorhamnetin-*O*-pentosyl-hexoside (*m*/*z* 609) and methylherbacetin-*O*-dihexoside (*m*/*z* 639) were detected in the raw BP in higher concentrations, while for raw BB, the most relevant were methyl herbacetin-*O*-dihexoside (*m*/*z* 639, isorhamnetin-*O*-pentosyl-hexoside (*m*/*z* 609), quercetin derivative (*m*/*z* 623) and the kaempferol-*O*-hexoside (*m*/*z* 447), as seen in Table 2. 

Concerning the BP-enriched bread, there is a clear change within the profile, with the disappearance of almost all the flavonoids, except quercetin-3-*O*-rhamnoside, which was detected with values in the range of 0.01 to 0.10 mg/g, as shown in Table 3. On the other hand, no phenolic compound was identified at all in bread with BB. The absence of phenolic compounds in the fortified bread with BP and BB could be attributed to their partial degradation by the temperature [32]. The same behavior was reported for biscuits fortified with BP, with a significant reduction in phenolic compounds after baking [15]. Another important point regarding this situation is that the phenolic compounds were found at lower concentrations in the raw bee products, so the dilution associated with the proportion of BP and BB added to bread may reduce the ability to quantify these compounds using this analytical approach.

Phenolamides were found in both raw bee products and cooked bread samples, which reflects their higher resistance to degradation under the baking conditions. Regarding the raw BP sample, the total amount of phenolamides was 14.82 mg/g. In particular, *N^1^*, *N^5^*, *N^10^*-tri-*p*-coumaroylspermidine (*m*/*z* 582) and its isomers (*m*/*z* 582), *N^1^*-*p*-coumaroyl-*N^5^*, *N^10^*-dicaffeoylspermidine (*m*/*z* 614) and tetracoumaroyl spermine (*m*/*z* 785) were detected at higher concentrations, a fact that was also observed for the BP-enriched bread. Nonetheless, the total amount of phenolamides in BP-enriched bread was significantly lower, ranging from 0.35 mg/g (BP 1%) to a maximum of 1.27 mg/g (BP 5%). 

For the BB raw sample, the total amount of phenolamides was 6.81 mg/g, which was less than half of that observed in BP. Despite that, the same phenolamide compounds, namely, *N^1^*, *N^5^*, *N^10^*-tricaffeoylspermidine (*m*/*z* 630), *N^1^*, *N^5^*, *N^10^*-tri-*p*-coumaroylspermidine (*m*/*z* 582) and tetracoumaroyl spermine (*m*/*z 785*), were detected at higher concentrations in both BP and BB-enriched bread. Although the total amount of bioactive compounds in raw BP was higher than the BB, when a comparison is made for the enriched bread, the total amount of bioactive compounds was higher in bread with BB. 

### 3.3. Antioxidant Activity

BP and BB are characterized by their antioxidant properties, which are often correlated with the phenolic compounds in their composition [28]. The antioxidant capacity of both BP and BB is dependent not only on the plant origin but also on the geographical origin and even on the collection time [28]. The antioxidant capacities of the raw BP, BB and enriched breads were measured by two different assays, DPPH radical scavenging and reducing power activity, and the obtained results are given in Figure 4a,b.

The radical scavenging activity of the BP sample was 0.20 ± 0.01 mg/mL, very close to that of BB. For the reducing power activity, raw BP revealed a better performance with 5.0 ± 0.01 mg GAE/g compared to 2.7 ± 0.20 mg GAE/g for BB. These results are in accordance with previously reported studies for BP and BB [12,27] and mean that the addition of them can lead to an increase in the antioxidant activity of bread.

Complementing the conventional bread with BP and BB at different ratios resulted in a significant increase in radical scavenging and reducing power activity compared with the control group, as seen in Figure 4c,d. According to the assays, while the DPPH radical scavenging of the control bread was 0.81 ± 0.01 mg/mL, introducing BP resulted in a decrease to 0.47 ± 0.01 mg/mL in BP 1% and to 0.33 ± 0.01 mg/mL in BP 5%. For the bread enriched with BB, DPPH radical scavenging values varied from 0.55 ± 0.01 mg/mL in BB 1% to 0.31 ± 0.01 mg/mL in BB 5%. The differences between the control and enriched bread were noticeable and statistically significant *(p* < 0.05). 

The same improvement was observed on the reduction power assay, with an increase in the ability to reduce the ion Fe^3+^ to Fe^2+^ as we add BP or BB to the bread recipe. Bread loaves enriched with BP ranged from 0.20 ± 0.01 to 0.30 ± 0.01 mg GAE/g, while BB-enriched bread results ranged from 0.11 ± 0.01 to 0.30 ± 0.01 mg GAE/g, both showing higher reducing power values than the control. Furthermore, in general, BP and BB-enriched bread samples showed quite similar reducing power activity profiles to each other, distorting the differences observed for the raw bee products.

In general, the antiradical and reduction ability observed for the enriched bread followed the same trend observed for the spectrophotometric evaluation of its phenolic content, as shown in Figure 3c,d, both increasing with the amount of bee product added to the enriched bread. Although we could expect some impact on the phenolic stability due to the baking conditions, the addition of bioactive ingredients in cooked products seems to enhance its antioxidant activity. Conte et al. observe that the addition of 5% of BP to gluten-free bread increased the polyphenol fraction of the soluble and insoluble extracts approximately by three times more than in the control bread, resulting in the enhancement of the DPPH and ABTS activity of around 10% [14]. The addition of BP to cookies, up to 15%, also revealed a significant increase in the antioxidant ability [17]; however, the authors highlighted that the stability of the pollen exine reduced the bioaccessibility of the phenolics compounds and so limited the antioxidant gain. Nevertheless, it should be noted that a complex series of chemical reactions such as enzymatic production of sugars, protein denaturation, yeast and enzyme inactivation, and Maillard reactions occur during bread production, and thus, significant changes may occur in the antioxidant activity by affecting the phenolic compounds and nutritional value of bread [48].

## 4. Conclusions

This study has revealed, for the first time, the phenolic profile and antioxidant performance of traditional bread fortified with BP and BB, at different concentrations. This fortification may create an effective approach to daily consumption of bee products. In addition, the thermostability of the bioactive compounds found in this functional bread was evaluated. The main phenolic compounds in the BP and BB were flavonol derivatives, together with phenolamides. Bread enriched with BP and BB showed both higher phenolic compound content and higher antioxidant activity compared to the control group. This indicates that both bee products retain the presence/activity; however, from the phenolic profile evolution, it was evident that the resistance of flavonoids and phenolamides to degradation, under the baking conditions, is different, with the latter evidencing higher thermostability.

Considering the results, this study confirms the positive impact in the bioactive composition and antioxidant activity of fortified bread with BP or BB, even in lower percentages. As a core product of daily consumption, cumulative intake can contribute significantly to the improvement of health conditions.

## Figures and Tables

**Figure 1 antioxidants-12-01691-f001:**
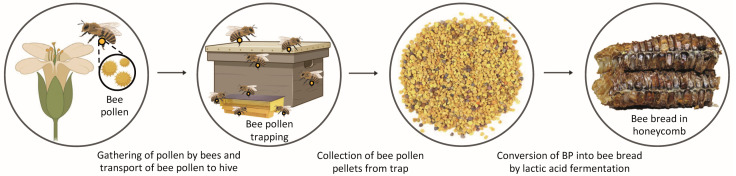
The diagram summarizes the process from gathering pollen grains from flowers by bees to forming bee bread.

**Figure 2 antioxidants-12-01691-f002:**
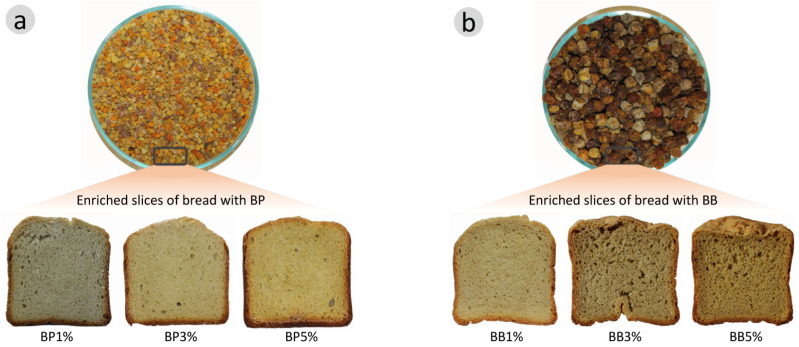
Slices of bread enriched with (**a**) bee pollen (BP) and (**b**) bee bread (BB).

**Figure 3 antioxidants-12-01691-f003:**
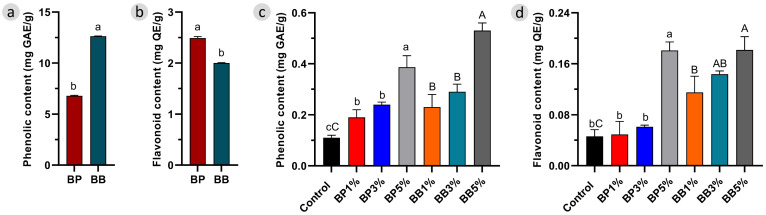
(**a**) Total phenolic content and (**b**) total flavonoid content of raw bee pollen (BP) and bee bread (BB), (**c**) total phenolic content and (**d**) total flavonoid content of enriched bread. Lowercase and uppercase letters denote statistical differences (*p* < 0.05) between BP and BB-enriched bread, respectively, compared to the control group.

**Figure 4 antioxidants-12-01691-f004:**
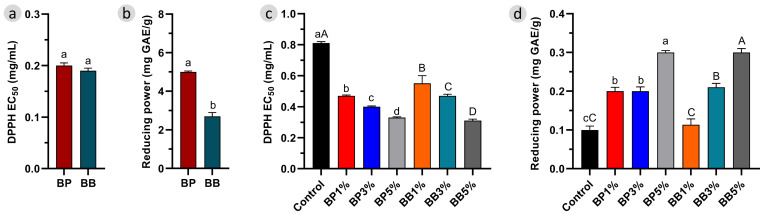
(**a**) DPPH radical scavenging and (**b**) reducing power activity of raw bee pollen (BP) and bee bread (BB), (**c**) DPPH radical scavenging, and (**d**) reducing power activity of enriched bread with BP and BB. Lowercase and uppercase letters denote statistical differences (*p* < 0.05) between BP and BB-enriched bread, respectively, compared to the control group.

**Table 1 antioxidants-12-01691-t001:** Formulations used to prepare the bread samples.

Ingredients	Control	BP 1%	BP 3%	BP 5%	BB 1%	BB 3%	BB 5%
Flour (g)	520.0	514.8	504.4	494.0	514.8	504.4	494.0
Water (mL)	300	300	300	300	300	300	300
Yeast (g)	5.0	5.0	5.0	5.0	5.0	5.0	5.0
Salt (g)	1.5	1.5	1.5	1.5	1.5	1.5	1.5
BP (g)	-	5.2	15.6	26.0	-	-	-
BB (g)	-	-	-	-	5.2	15.6	26.0
Total	826.5	826.5	826.5	826.5	826.5	826.5	826.5

BP: bee pollen and BB: bee bread.

**Table 3 antioxidants-12-01691-t003:** Phenolic and phenolamide compounds of the enriched bread with BP and BB (mg/g).

Peak *	Proposed Compound	BP 1%	BP 3%	BP 5%	BB 1%	BB 3%	BB 5%
13	Quercetin-3-*O*-rhamnoside	0.01 ± 0.00	0.09 ± 0.00	0.10 ± 0.00	ND	ND	ND
18	*N^1^*, *N^5^*, *N^10^*-tricaffeoylspermidine	0.01 ± 0.00	0.04 ± 0.00	0.06 ± 0.00	0.01 ± 0.00	0.11 ± 0.00	ND
20	*N^1^*, *N^5^*, *N^10^*-tricaffeoylspermidine (isomer)	ND	ND	ND	0.03 ± 0.00	0.34 ± 0.00	0.36 ± 0.00
23	*N^1^*-*p*-coumaroyl-*N^5^*, *N^10^*-dicaffeoylspermidine (isomer)	ND	ND	ND	ND	0.01 ± 0.00	0.01 ± 0.00
24	*N^1^*-*p*-coumaroyl-*N^5^*, *N^10^*-dicaffeoylspermidine (isomer)	0.01 ± 0.00	0.02 ± 0.00	0.10 ± 0.00	ND	0.10 ± 0.00	0.08 ± 0.00
25	*N^1^*-feruloyl-*N^5^*, *N^10^*-dicaffeoylspermidine (isomer)	ND	ND	ND	ND	ND	0.01 ± 0.00
26	*N^1^*, *N^5^*-di-*p*-coumaroyl-*N^10^*-caffeoylspermidine	0.01 ± 0.00	0.01 ± 0.00	0.08 ± 0.00	0.02 ± 0.00	0.19 ± 0.00	0.31 ± 0.00
27	*N^1^*, *N^5^*, *N^10^*-tri-*p*-coumaroylspermidine	0.05 ± 0.00	0.09 ± 0.00	0.18 ± 0.00	ND	0.13 ± 0.00	0.15 ± 0.00
29	*N^1^*, *N^5^*, *N^10^*-tri-*p*-coumaroylspermidine (isomer)	0.02 ± 0.00	0.04 ± 0.00	0.04 ± 0.00	0.03 ± 0.00	0.44 ± 0.00	0.46 ± 0.00
30	*N^1^*, *N^5^*, *N^10^*-tri-*p*-coumaroylspermidine (isomer)	0.15 ± 0.00	0.10 ± 0.00	0.37 ± 0.00	0.04 ± 0.00	0.41 ± 0.00	0.49 ± 0.00
31	*N^1^*, *N*^5^, *N^10^*-tri-*p*-coumaroylspermidine (isomer)	ND	ND	ND	0.01 ± 0.00	0.23 ± 0.00	0.35 ± 0.00
32	*N^1^*, *N^5^*, *N^10^*-tri-*p*-coumaroylspermidine (isomer)	ND	ND	ND	0.15 ± 0.00	0.85 ± 0.10	1.69 ± 0.00
33	Tetracoumaroyl spermine	0.07 ± 0.00	0.02 ± 0.00	0.01 ± 0.00	ND	0.06 ± 0.00	0.04 ± 0.00
34	Feruloyl dicoumaroyl spermidine	0.01 ± 0.00	ND	0.04 ± 0.00	ND	ND	ND
36	Tetracoumaroyl spermine (isomer)	ND	0.11 ± 0.00	0.19 ± 0.00	ND	ND	ND
37	Tetracoumaroyl spermine (isomer)	0.02 ± 0.00	0.03 ± 0.00	0.07 ± 0.00	ND	ND	ND
38	Tetracoumaroyl spermine (isomer)	ND	0.04 ± 0.00	0.13 ± 0.00	ND	0.09 ± 0.00	0.07 ± 0.00
Total phenolic compounds (mg/g)		0.01 ± 0.00	0.09 ± 0.00	0.10 ± 0.00	–	–	–
Total phenolamides (mg/g)		0.35	0.50	1.27	0.29	2.96	4.17

BP: bee pollen and BB: bee bread. Values are expressed as mg of each compound/g sample. ND = not detected. * Peak number according to Table 2.

## Data Availability

Not applicable.

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
