# Peer review of "Thermal Stability and Antioxidant Activity of Bioactive Compounds in Bread Enriched with Bee Pollen and Bee Bread"

_antioxidants, 2023, doi:10.3390/antiox12091691_

Round 1

Reviewer 1 Report

The study design is sound and well-executed. And the manuscript is well organized, sufficiently but not overly brief, and the data are well-presented. The conclusions are well in line with the results. Overall, a timely and high quality report.

The manuscript could use minor editing for English language proficiency, especially in the abstract and the introduction. The results & discussion and conclusion sections could go without further editing.

Author Response

  1. The study design is sound and well-executed. And the manuscript is well organized, sufficiently but not overly brief, and the data are well-presented. The conclusions are well in line with the results. Overall, a timely and high-quality report.

The manuscript could use minor editing for English language proficiency, especially in the abstract and the introduction. The results & discussion and conclusion sections could go without further editing.

Our response: The entire manuscript was checked for language improvement, but particular attention was given to the abstract and introduction.

Reviewer 2 Report

The submitted manuscrypt seems to be quite an original work, which in my opinion is worth publishing in antioxidants. The authors made a valuable comparison of the stability of polyphenolic compounds in bread under the influence of temperature. I also believe that in addition to the main goal, a quite interesting new product from the fortified food sector has been proposed.

However, I have a few technical comments that I hope will help improve the manuscript.

First of all, I think that in the introduction there is too much information about theoretical issues related to functional foods. I believe that much has already been said on this topic in numerous scientific studies. Instead, the authors should focus more on the issues related to the use of bee products and their potential in the production of fortified foods.

Paragraph 2.2 please specify which exact province the material came from. I suggest changing the topic of paragraph 2.5 to a more general one, e.g. "Spetrophotometric determination of phenolic compounds"

Paragraph 2.6 lines 152-154 are unclear. Was the extract from the previous protocol taken or was the extraction performed again? Were the samples dissolved or diluted?

Figure 2. The whole drawing has been divided into 2 parts a) and b) while in the description there is a reference to part c) ?

In the discussion, the authors refer to the differences in the content of polyphenolic compounds in pollen and bee bread, referring to the differences in origin and environmental conditions. I suggest taking into account the nature of the raw materials used. Bee bread is a product processed by bees where it undergoes lactic fermentation, so this may also have an effect.

In table 3,  also proposes to add the total contents similarly to what was done in table 2.

Author Response

The manuscript is now revised taking into consideration the proposals of the reviewers. We also highlight in red all the changes made in the manuscript.

We would like to thank the reviewers for their valuable comments, suggestions, and contributions to this study.

Reviewer 2:

"The submitted manuscript seems to be quite an original work, which in my opinion is worth publishing in antioxidants. The authors made a valuable comparison of the stability of polyphenolic compounds in bread under the influence of temperature. I also believe that in addition to the main goal, a quite interesting new product from the fortified food sector has been proposed.

However, I have a few technical comments that I hope will help improve the manuscript."

  1. First of all, I think that in the introduction there is too much information about theoretical issues related to functional foods. I believe that much has already been said on this topic in numerous scientific studies. Instead, the authors should focus more on the issues related to the use of bee products and their potential in the production of fortified foods.

Our response: Some introductory information on functional food was removed from the introduction as recommended. A specific paragraph was added referring to food products incorporated with bee products at the suggestion of the reviewer.

  1. Paragraph 2.2 please specify which exact province the material came from. I suggest changing the topic of paragraph 2.5 to a more general one, e.g. "Spectrophotometric determination of phenolic compounds".

Our response: Changes were made in the text following the reviewer suggestion.

  1. Paragraph 2.6 lines 152-154 are unclear. Was the extract from the previous protocol taken or was the extraction performed again? Were the samples dissolved or diluted?

Our response: Clarification was made to address the reviewer concerns.

  1. Figure 2. The whole drawing has been divided into 2 parts a) and b) while in the description there is a reference to part c) ?

Our response: A correction was made in the Figure 2 caption following the reviewer suggestion.

  1. In the discussion, the authors refer to the differences in the content of polyphenolic compounds in pollen and bee bread, referring to the differences in origin and environmental conditions. I suggest taking into account the nature of the raw materials used. Bee bread is a product processed by bees where it undergoes lactic fermentation, so this may also have an effect.

Our response: Thanks for the reviewer's suggestion. We cannot exclude that differences may also arise from the lactic fermentation of BB, so we add also that possibility within the discussion.

  1. In table 3, also proposes to add the total contents similarly to what was done in table 2.

Our response: Changes were made in Table 3 following the reviewer suggestion.

Reviewer 3 Report

The authors tried to show the effect to heat treatment on the bioactive activity of bee pollen and bee bread. Even though there are some effective data and conclusions from the manuscript, most results are predictable. Here are some recommendations:

1. Line 31, select The. Type errors reflect the carelessness of the author.

2. The main reason for the decision is due to the citation 24 (Ertosun, S., et al., The impact of bee products incorporation on the processing properties, nutritional value, sensory acceptance, and 417 microbial stability of bread. Journal of Food Measurement Characterization, Submitted.) 

I have never seen this type of citation of the submitted manuscript. Based on your submitted study, you should include the antioxidant part in the before-submitted manuscript, not take it out to organize an incomplete study. 

3.  I'm confused about the DPPH and reducing power results.  The author stated that they used sample extract to test the antioxidant activity. But how do they unify the bioactive concentration? Based on the concentration of BP and BB in bread? 

The concentration of bioactive changed a lot from BP or BB to bread. Even though the BP or BB was added at 1%-5%, after the baking process, the water evaporated, and the concentration of BP or BB is not the content before. Then, during the extraction, the concentration of the bioactive also changed. 

Author Response

The manuscript is now revised taking into consideration the proposals of the reviewers.  We also highlight in red all the changes made in the manuscript.

We would like to thank the reviewers for their valuable comments, suggestions, and contributions to this study.

Reviewer #3:

"The authors tried to show the effect to heat treatment on the bioactive activity of bee pollen and bee bread. Even though there are some effective data and conclusions from the manuscript, most results are predictable. Here are some recommendations:"

  1. Line 31, select The. Type errors reflect the carelessness of the author.

Our response: Thanks to the reviewer comment. A complete revision on the manuscript was made in order to avoid any misspelling.

  1. The main reason for the decision is due to the citation 24 (Ertosun, S., et al., The impact of bee products incorporation on the processing properties, nutritional value, sensory acceptance, and 417 microbial stability of bread. Journal of Food Measurement Characterization, Submitted.)

I have never seen this type of citation of the submitted manuscript. Based on your submitted study, you should include the antioxidant part in the before-submitted manuscript, not take it out to organize an incomplete study.

Our response: The work concept on enriched bread with bee pollen and bee bread was organized in two different research stages: the first oriented for the physical, sensorial, microbial and nutritional properties of the functionalized breads and then, a specific stage focused on the bioactivity gain of these new food products. The amount and quality of the scientific results disabled the possibility of combining all the information in only one manuscript, so the authors organized two different publications, accordingly to the above mentioned stages, and considering that both manuscript results are independent. The first publication was submitted in march 2023, and the second four months’ latter, after submission of the revision version of the first manuscript. Unfortunately, not all publishers performed as Antioxidants in respect to the time response, and so the publication of the first manuscript is delayed. Nevertheless, the citation was made following the guidelines of the journal for manuscripts under submission.

  1. I'm confused about the DPPH and reducing power results. The author stated that they used sample extract to test the antioxidant activity. But how do they unify the bioactive concentration? Based on the concentration of BP and BB in bread?

The concentration of bioactive changed a lot from BP or BB to bread. Even though the BP or BB was added at 1%-5%, after the baking process, the water evaporated, and the concentration of BP or BB is not the content before. Then, during the extraction, the concentration of the bioactive also changed.

Our response: Thank you for the comment. Measuring the bioactive compound contents and antioxidant activities of raw BP and BB extracts gave us an insight into the content of the tested samples, however, these values were not used to compare with bread fortified with bee products. The control bread (without BP or BB) and bread containing different percentages of BP and BB were freeze-dried after baking, thus removing water, and then subjected to an extraction procedure using the same weight of the sample. So the antioxidant results for all breads are expressed and compared in dry sample weight. Nevertheless, and considering the question raised by the reviewer, we made some improvements in the section ''Materials and Methods'' to clarify this issue.

Round 2

Reviewer 3 Report

The revised version is acceptable.